Comparing the effects of empagliflozin and liraglutide on lipid metabolism and intestinal microflora in diabetic mice

Yang Qiong 1
Deng Ling 1
Feng Changmei 1
http://orcid.org/0009-0004-5928-8560 Wen Jianghua 2 wen_j_h@163.com
1 Nanxishan Hospital of Guangxi Zhuang Autonomous Region , Guilin , China
2 Jiangmen Central Hospital , Jiangmen , China
Oliveira Sonia
Electronic publication date: 2024 Mar 15
Publication date: 2024
Volume: 12
Electronic Location ID: e17055
Received 2023 Sep 18; Accepted 2024 Feb 14
Copyright: © 2024 Yang et al.
Copyright year: 2024
Copyright holder: Yang et al.
License: This is an open access article distributed under the terms of the Creative Commons Attribution License, which permits unrestricted use, distribution, reproduction and adaptation in any medium and for any purpose provided that it is properly attributed. For attribution, the original author(s), title, publication source (PeerJ) and either DOI or URL of the article must be cited.
License URL: https://creativecommons.org/licenses/by/4.0/

Keywords: Diabetic nephropathy, Lipid metabolism, Intestinal flora, Liraglutide, Empagliflozin

Funding: Guangxi Natural Science Foundation 2020GXNSFBA238001 National Natural Science Foundation of China 82060671 This research was supported by Guangxi Natural Science Foundation under Grant No. 2020GXNSFBA238001 and the National Natural Science Foundation of China under Grant 82060671. The funders had no role in study design, data collection and analysis, decision to publish, or preparation of the manuscript.

==============================
Background and Objectives

Recent studies have shown that the imbalance of intestinal flora is related to the occurrence and progression of diabetic nephropathy (DN) and can affect lipid metabolism. Sodium-dependent glucose transporters 2 (SGLT2) inhibitor and glucagon-like peptide-1 (GLP-1) receptor agonist are commonly used hypoglycemic drugs and have excellent renal safety. The purpose of this study was to compare the protective effects of empagliflozin and liraglutide on kidneys, lipid metabolism, and intestinal microbiota in diabetic mice.

Methods

We established a mouse model of type two diabetes by feeding rats a high-fat diet (HFD) followed by an intraperitoneal injection of STZ. The mice were randomly divided into groups: normal control (NC), diabetic model (DM), liraglutide treatment (LirT), empagliflozin treatment (EmpT), and liraglutide combined with empagliflozin treatment (Emp&LirT) groups. Blood glucose, lipids, creatinine, and uric acid, as well as urinary nitrogen and albumin levels were measured. The renal tissues were subjected to HE, PAS and Masson’s staining. These parameters were used to evaluate renal function and histopathological changes in mice. Mice feces were also collected for 16sRNA sequencing to analyze the composition of the intestinal flora.

Results

All the indexes related to renal function were significantly improved after treatment with drugs. With respect to lipid metabolism, both drugs significantly decreased the serum triglyceride levels in diabetic mice, but the effect of liraglutide on reducing serum cholesterol was better than that of empagliflozin. However, empagliflozin had a better effect on the reduction of low-density lipoproteins (LDL). The two drugs had different effects on intestinal flora. At the phylum level, empagliflozin significantly reduced the ratio of Firmicutes to Bacteroidota, but no effect was seen with liraglutide. At the genus level, both of them decreased the number of Helicobacter and increased the number of Lactobacillus. Empagliflozin also significantly increased the abundance of Muribaculaceae, Muribaculum, Olsenella, and Odoribacter, while liraglutide significantly increased that of Ruminococcus.

Conclusion

Liraglutide and empagliflozin were both able to improve diabetes-related renal injury. However, the ability of empagliflozin to reduce LDL was better compared to liraglutide. In addition, their effects on the intestine bacterial flora were significantly different.

Introduction

Diabetic nephropathy (DN) is one of the chronic complications of diabetic patients. Its natural course includes glomerular hyperfiltration, progressive proteinuria, glomerular filtration rate (GFR) decline, and finally, end-stage renal disease (ESRD). Chronic kidney disease (CKD) is the main cause of these symptoms. Epidemiological surveys in recent years have shown that 40% of diabetic patients have DN (Alicic, Rooney & Tuttle, 2017). The pathogenesis of DN is complex, and there is increasing evidence linking it to abnormal lipid metabolism (Nagai et al., 2022) and ectopic renal lipid deposition (EFD) (Huang et al., 2023; Yang et al., 2018), which are both important risk factors for this condition. EFD induces inflammation and fibrosis of the renal parenchyma and interstitium, leading to glomerular epithelial cell damage and dysfunction and further promoting the progression of DN (Huang et al., 2023; Zhou et al., 2023). Regulating lipid metabolism may be an effective way to alleviate the progression of DN.

Recent studies have shown that metabolites produced by commensal gut microbes can affect many metabolic pathways, particularly lipid metabolism, through recognition by the immune system (Brown, Clardy & Xavier, 2023). Enterobacter enterica is highly colonized in the gut of obese and hyperglycemic patients and contributes to diet-induced obesity through the production of long-chain fatty acids (Takeuchi et al., 2023). However, Bacteroidota faecalis and Clostridium spp. have been shown to significantly enrich in fecal samples of DN patients (Zhang et al., 2022). In addition, significant correlations have been shown with clinical indicators of lipid and glucose metabolism as well as renal function (Zhang et al., 2022). Supplementation with beneficial bacteria can inhibit the bodies’ inflammatory and oxidative stress responses, thus playing a beneficial role in patients with ESRD (Han et al., 2023; Soleimani et al., 2017; Li et al., 2020). This suggests that improving diabetes-related nephropathy through the “gut-kidney axis” may be a potential treatment of DN.

Glucagon-like peptide 1 (GLP-1) receptor agonists and sodium-glucose cotransporter 2 (SGLT2) inhibitors have been shown to be effective in the treatment of type 2 diabetes while improving the end-stage outcome of DN (Zhang et al., 2023; Chen et al., 2023; Prattichizzo, de Candia & Ceriello, 2021). However, the mechanisms of action and therefore the impact of these two types of drugs on diabetic renal outcomes are different (Hocher & Tsuprykov, 2017). Many studies have shown that the regulation of intestinal flora is an important part of the potential mechanism of liraglutide and empagliflozin in the treatment of type two diabetes as well as kidney protection in diabetic mice.

Therefore, in this study, empagliflozin and liraglutide were administered to diabetic mice, and the curative effects of each group of intervention measures and the effects on the intestinal flora, lipid metabolism and renal pathological damage of the mice were observed. We also investigated the theoretical basis for the renal protection of empagliflozin and liraglutide.

Materials and Methods

Animals

Six-week-old male C57BL/6J mice, weighing 18–20 g, were purchased from Beijing Weitong Lihua Experimental Animal Technology Co., Ltd. The experimental protocols of all mice were approved by the Animal Experiment Ethics Committee of Guangxi Medical University (No. 20202014). Mice were raised in a standard SPF-level experimental animal breeding environment with a temperature of 26 °C, humidity of 60%, and alternating light and dark illumination for 12 h/12 h. The mice were fed a free-flowing diet supplemented with food (two to three times per week) by a dedicated laboratory breeder. To ensure enough space for the mice, we raised four to six mice in each cage, and the cage size was suitable was an iron grid for the mice to climb. As previously described (Mulvihill et al., 2016), the diabetic model was induced by a combination of a high-fat diet (HFD) and intraperitoneal injection of streptozotocin.

The diabetic group (N = 40) was fed with a HFD (45% kcal fat, 20% protein, 35% kcal carbohydrate, 0.2% cholesterol) for 12 weeks, and a single dose of STZ was injected intraperitoneally at the 13th week (90 mg/kg ip, freshly prepared 0.1 mmol/L sodium citrate solution, pH5.5), once a day for 5 consecutive days. One week later, fasting blood glucose (FBG) was monitored (twice on different days). If the FBG was greater than or equal to 11.1 mmol/L, the modeling was deemed successful (Wang et al., 2022). The normal control group (control, N = 10) was fed with a normal diet for 12 weeks, and the same volume of sodium citrate solution was injected intraperitoneally starting from the 13th week for five consecutive days, and the FBG was measured one week later. The diabetic mice were divided into four groups (randomly assigned according to their ear tag numbers): diabetic model (DM), liraglutide treatment (LirT), empagliflozin treatment (EmpT) and empagliflozin combined with liraglutide treatment (Emp&LirT) groups. The therapeutic doses of liraglutide and empagliflozin were 0.250 and 25 mg/kg/d (Chen et al., 2013; Herat et al., 2020) and lasted for 8 weeks. At the termination of treatments urine, feces, blood and kidney tissues were collected from mice for subsequent assessments.

All animal experiments were conducted in accordance with the national standards of the People’s Republic of China ethical review guidelines for experimental animal welfare (GB/T35892-2018). In the experiment, we considered early termination when significant weight loss (rapid onset of 15–20% of body weight), weakness, dying, or organ failure were observed; otherwise, the experiment was completed when the expected goal was reached.

Mice FBC

The mice fasted for 6 h, and the blood glucose levels of samples obtained from the tail veins were measured using an ACCU-CHECK blood glucose meter (Roche Diagnostics, Mannheim, Germany). The method for collection of blood was to cut off about 1 mm from the tip of the tail after it was disinfected with an alcohol cotton ball. The first drop of venous blood was discarded, and the second drop of blood (0.1 ml) was collected. The tail of the mouse was then pressed for 1 min with a dry cotton ball to stop the bleeding.

Biochemical measurements of sera and urine

The serum samples were obtained by taking blood from the back of the eyeballs. The mice were intraperitoneally injected with 10% chloral hydrate (0.1 mL/10 g weight). The left eyeballs of the mice were removed with hemostatic forceps and blood samples were collected under deep anesthesia (skin pinch reaction: the mouse skin was pinched with toothed forceps, no obvious reaction). After the blood was kept at room temperature for 4 h, the cells and serum were separated by centrifugation (3,000 rpm, 10 min). The urine samples of mice were collected using a special metabolic cage (24 h) and supernatants obtained after centrifugation were used for measurements (1,000 rpm, 5 min). All samples are processed and stored in an ultra-low temperature refrigerator at −80 °C. Urine creatinine and albumin of mice were measured using mouse ELISA kits (cloud-clone corp. Wuhan), and other indexes were assessed in an automatic biochemical analyzer (URIT-500B).

Renal tissue staining

According to the above method (experimental method 3), the mice were anesthetized, blood was taken, and mice were killed by cervical dislocation. The kidney tissues of mice were collected, fixed with paraformaldehyde, dehydrated, embedded, and sectioned by standard procedure. Renal pathological changes were evaluated by HE, PAS and Masson staining, and the expression of the related proteins was evaluated by immunohistochemical staining. The images were obtained under a pathological microscope and semi-quantitative analysis was realized with image J software.

HE staining

Dewaxing and hydration: The paraffin sections were immersed in sequence in environmental friendly dewaxing Transparent Liquid I for 20 min—environmental friendly dewaxing Transparent Liquid II for 20 min—anhydrous ethanol I for 5 min—anhydrous ethanol II for 5 min—75% Ethyl alcohol for 5 min, and then rinsed with tap water.

Rewarming and fixing: The frozen sections were removed from the −20 °C refrigerator and restored to room temperature, fixed with tissue fixating solution for 15 min, and then rinsed with running water.

Hematoxylin staining: Put sections into Hematoxylin solution for 3–5 min, rinse with tap water. Then treat the section with Hematoxylin differentiation solution, rinse with tap water. Treat the section with Hematoxylin bluing solution, rinse with tap water.

Eosin staining: Place the sections in sequence in 85% ethanol for 5 min—95% ethanol for 5 min—Eosin dye for 5 min.

Dehydration and sealing: Put the sections into absolute ethanol I for 5 min—absolute ethanol II for 5 min—absolute ethanol III for 5 min—xylene I for 5 min—xylene II for 5 min, sealing with neutral gum.

PAS staining

Dewaxing and hydration: The paraffin sections were immersed in sequence in Environmental Friendly Dewaxing Transparent Liquid I for 20 min—Environmental Friendly Dewaxing Transparent Liquid II for 20 min—anhydrous ethanol I for 5 min—anhydrous ethanol II for 5 min—75% Ethyl alcohol for 5 min, and then rinsed with tap water.

Rewarming and fixing: The frozen sections were removed from the −20 °C refrigerator and restored to room temperature, fixed with tissue fixating solution for 15 min, and then rinsed with running water.

Stain sections with PAS dye solution B for 10–15 min, rinse with tap water, and rinse twice with distilled water.

Stain with PAS dye solution A for 25–30 min in the dark, rinse for 5 min.

Then stain sections with PAS dye solution C for 30 s, and rinse with tap water. Treat the slices with hydrochloric acid solution and ammonia, each step required washing with water.

Dehydration and sealing: Put the sections into absolute ethanol I for 5 min-absolute ethanol II for 5 min-absolute ethanol III for 5 min—xylene I for 5 min—xylene II for 5 min, sealing with neutral gum.

Microscope inspection, image acquisition and analysis.

Masson staining

Dewaxing and hydration: The paraffin sections were immersed in sequence in Environmental Friendly Dewaxing Transparent Liquid I for 20 min—Environmental Friendly Dewaxing Transparent Liquid II for 20 min—anhydrous ethanol I for 5 min—anhydrous ethanol II for 5 min—75% ethyl alcohol for 5 min, and then rinsed with tap water.

Rewarming and fixing: The frozen sections were removed from the −20 °C refrigerator and restored to room temperature, fixed with tissue fixating solution for 15 min, and then rinsed with running water.

The slices were soaked in Masson A overnight, rinse with tap water.

Masson B and Masson C were prepared into Masson solution according to the ratio of 1:1. Then stain with Masson solution for 1 min, rinse with tap water. Differentiate with 1% hydrochloric acid alcohol for several seconds, rinse with tap water.

Soak the slices in Masson D for 6 min, rinse with tap water;

Masson E for 1 min;

Do not wash the slides; slightly drain directly into Masson F for 2–30 s.

Rinse the slices with 1% glacial acetic acid and then dehydration with two cup of anhydrous ethanol.

Clearing and sealing: slides were soaked in 100% ethanol for 5 min; Xylene for 5 min; finally sealed with neutral gum.

Microscope inspection, image acquisition and analysis.

SDNA sequencing and analysis

After the experimental treatment, the fresh fecal particles of mice were collected and stored at −80 °C daily until DNA extraction (according to the experimental requirements, each group of mice was to provide five samples, which we randomly selected for testing). The total DNA was extracted from 150 mg of the rectal contents. DNA from different samples was extracted using the CTAB according to manufacturer’s instructions (RNA from total samples was isolated and purified with TRIzol (15596018; Thermo Fisher, Waltham, MA, USA) according to the operation protocol provided by the manufacturer.). The reagent which was designed to uncover DNA from trace amounts of sample has been shown to be effective for the preparation of DNA of most bacteria. Nuclear-free water was used for blank. The total DNA was eluted in 50 μL of Elution buffer and storedat −80 °C until measurement in the PCR by LC-Bio Technology Co., Ltd, Hang Zhou, Zhejiang Province, China. An appropriate amount of the sample was centrifuged and diluted to 1 ng of sample/μL using sterile water. The diluted genomic DNA was used as a template for PCR amplification using the primers 341F (5′-CCTACGGGNGGNGGCWGCAGMur-3′), and 805R (5′-GACTACHVGGGTATCTAATCC-3′). After purification and quantification of PCR products, the library construction was evaluated by using an Illumina (Kapa Biosciences, Woburn, MA, USA) library construction kit. Then 2 × 250 bp double-terminal sequencing was carried out with NovaSeq 6000 sequencer, and the corresponding NovaSeq 6000 SP Reagent Kit (500 cycles).

According to the barcoded and PCR primer sequences, the sample data were separated from the offline data, and the reads part of the sample was spliced using FLASH after truncating the sequences. The tags quality control process of Qiime was used, and these were compared with the database (Gold database, http://drive5.com/uchime/uchime_download.html) to detect the chimeric sequences. These were removed, and qiime dada2 denoise-paired was used in order to recall DADA2 for length filtering and denoising. The ASV (feature) feature sequence and the ASV (feature) abundance table were obtained, and the singletons ASVs were removed (that is, ASV (feature) with a total sequence of only one in all the samples, the default operation).

Based on the obtained ASV (feature) feature sequence and ASV (feature) abundance table, alpha multiplicity and beta diversity analyses were carried out. According to the ASV (feature) sequence file, the SILV A (Release 138, https://www.arb-silva.de/documentation/release138/) database was used to annotate the species with the NT-16S database, and the abundance of each species in each sample was counted according to the ASV (feature) abundance table. The confidence threshold of the annotation was set to P < 0.7. Based on the statistical information with respect to species abundance, the Kruskal-Wallis test was used to analyze the differences among the groups. This part of the experiment was entrusted to Lianchuan Biotechnology Co. Ltd.

Statistical analysis

The results were expressed as means ± SDs. SPSSv20.0 and Prism8.4.3GraphPad software packages were used for statistical analysis. Single factor analysis of variance (ANOVA) was used for statistical comparisons among the groups. P < 0.05 was considered to be significantly different.

Results

Effects of empagliflozin and liraglutide on renal function in diabetic mice

The FBG of mice induced by HFD plus STZ increased significantly, so we selected the mice whose FBG was higher than 11.1 mmol/L for follow-up experiments (mice were in good condition and follow-up experiments could be carried out without euthanasia). After 8 weeks of treatment, no mice died. The blood glucose of the mice in the drug treatment group decreased significantly, and compared with empagliflozin, liraglutide showed a greater hypoglycemic advantage (Fig. 1A). The biochemical results from the serum and urine of mice showed that both empagliflozin and liraglutide improved the renal function of diabetic mice. Empagliflozin significantly decreased serum urea nitrogen in diabetic mice (P < 0.01), and empagliflozin and treatment combined with liraglutide also reduced the levels of serum urea nitrogen to some extent (Fig. 1B). Both of them could significantly reduce the levels of serum creatinine, and of the two drugs, liraglutide showed a better effect (Fig. 1C). The results of urinary protein showed that both of them could significantly reduce the urinary protein excretion of diabetic mice, and the urinary protein excretion seen with empagliflozin was significantly lower than that of liraglutide (P < 0.0001). However, the combination of the two treatments weakened the effect of empagliflozin on reducing urinary protein levels (Fig. 1D).

Figure 1 Protective effects of Liraglutide and empagliflozin on the kidneys of diabetic mice.

(A) Fasting blood glucose of mice in each group; (B) serum urea nitrogen levels of mice in each group; (C) serum creatinine levels of mice in each group; (D) urinary albumin excretion of mice in each group; (E) kidney pathological staining: HE, PAS, Masson staining. White arrow indicates dilated renal tubules, black arrow shows PAS positive staining (red), indicating thickened basement membrane; Green arrows are Masson-positive staining (blue), indicating fibrotic tissue. (F) Masson-stained area of the mouse kidney as a percentage of fibrotic tissue. magnification: 200×, scale bar: 20 μm. *P < 0.05, **P < 0.01, ***<0.001, ****P < 0.0001.

In order to further evaluate the pathological changes of the mouse kidney, we performed HE, PAS and Masson staining (Fig. 1E). HE staining showed that compared with the control group, there were a large number of inflammatory cell infiltration, disordered cell arrangement, glomerular hypertrophy, renal tubular dilatation and renal interstitial edema in the DM group. Both liraglutide and empagliflozin treatment reduced inflammatory cell infiltration to some extent and improved renal tubular dilatation, but the effects was not marked (Fig. 1E). PAS staining showed that compared with the control group, glomerular basement membrane thickening, balloon space widening, slight glomerular hypertrophy and occasional tubular atrophy were observed in the DM group. In addition, liraglutide, empagliflozin and combined treatment significantly improved glomerular basement membrane thickening and mesangial dilatation, and there were significant differences among the three groups (Fig. 1E). Masson staining showed that renal tubulointerstitial fibrosis occurred in diabetic mice compared with the control group Liraglutide and empagliflozin treatment reduced the fibrosis area, and the combined therapy also showed obvious advantages in inhibiting renal fibrosis compared with monotherapy (Fig. 1F).

Effects of empagliflozin and liraglutide on lipid metabolism in diabetic mice

We measured the lipid content in the sera of mice. The results showed that the levels of serum triglyceride, cholesterol and low-density lipoprotein (LDL) in diabetic mice were significantly higher than those in the control group. Liraglutide (P < 0.01) and empagliflozin (P < 0.001) could significantly reduce the levels of serum triglycerides in diabetic mice, especially in the Emp&LirT group (P < 0.001) (Fig. 2A). However, liraglutide and empagliflozin and their combination therapy could not significantly reduce serum cholesterol or LDL. Additionally, our data suggested that compared with the empagliflozin, liraglutide may still be better than in reducing cholesterol levels (6.1 mmol/LVS6.64 mmol/L) (Fig. 2B). With respect to the reduction of LDL, the effect of empagliflozin may be better (0.51 mmol/LVS0.37 mmol/L) (Fig. 2C). However, the combined therapy could not significantly reduce the levels of serum lipids in diabetic mice.

Figure 2 Effect of liraglutide and empagliflozin on serum lipid levels in mice.

(A) The serum triglyceride level; (B) the serum cholesterol level; (C) the serum low-density lipoprotein level. *P < 0.05, **P < 0.01, ***<0.001, ****P < 0.0001.

Effects of empagliflozin and liraglutide on intestinal microflora diversity in diabetic mice

We analyzed the average number of species in each group by using the Chao1 index (Fig. 3A). The result showed that the microbial α diversity of intestinal flora in the DM group was lower than that in control group (P < 0.01). The Chao1 index further decreased after treatment with liraglutide and empagliflozin, but there was no significant difference. The Chao1 index of the control, DM, LirT, EmpT and Emp&LirT groups were 627.67, 486.54, 431.03, 434.66 and 435.67, respectively.

Figure 3 Effect of liraglutide and empagliflozin on microbial diversity in the gut of mice.

(A) Alpha diversity of the gut microbiota in each group was analyzed by Chao1 Index; (B) beta diversity of the gut microbiota in each group was analyzed by PCOA component. R = 0.9930, P = 0.001.

Principle coordinate analysis (PCoA) is used to reflect the similarities of microbial structure across groups of samples, and the distance between samples can represent the similarity of microbial clustering, and the closer the distance between points, the higher the similarity of two samples, the greater the distance, the greater the difference. The results showed that the microbial clustering of the samples in the same group was very high, and the different treatment methods contributed to different microbial community characteristics. There was a significant difference between the normal control and diabetic samples, and both liraglutide and empagliflozin and combination therapy led to different differences; Among them, the sample of empagliflozin group was similar to that of the diabetic group, but samples from the combination group were more similar to those from the control group. In addition, liraglutide had a greater effect on the structure of the gut microbes than did empagliflozin (Fig. 3B).

Effects of empagliflozin and liraglutide on the composition of intestinal microflora in diabetic mice

At the phyla level, Fimicutes and Bacteroidota are the microbiologically dominant phylum in mice intestinal (Fig. 4A). Compared with the control group, the intestinal Firmicutes in the DM group significantly increased (31.84% and 61.59%, respectively; P = 0.048). Bacteroidota, Campylobacterota and Proteobacteria decreased and the ratio of Fimicutes and Bacteroidota (F/B) increased (F/B = 7.45). Liraglutide did not improve the imbalance of F/B, but significantly reduced the abundance of Desulfobacterota (P = 0.005), while empagliflozin improved the structural imbalance of the dominant flora (F/B = 2.04). The effect of combination therapy on the dominant bacteria was similar to that of the LirT group (F/B = 7.55), while the abundance of Verrucomicrobiota significantly increased from 0.12% to 3.81%.

Figure 4 Effect of liraglutide and empagliflozin on the structure of mouse intestinal flora.

(A) Column stacking plot of relative abundance of mouse intestinal flora at phylum level; (B) cluster heatmap of the mouse gut microbiota in the top 30 relative abundance at the genus level; (C) histogram of the relative abundance of the mouse gut microbiota at the genus level.

At the genus level, we selected the top 30 microbes for analysis. The cluster map showed that the DM and EmpT groups were in the same small branch together with the control group, while the LirT and Emp&LirT groups belonged to another small branch (Fig. 4B). Relative abundance analysis showed that Muribaculaceae the most dominant genus in the control group (Fig. 4C), followed by Lachnospiraceae, Muribaculum, Helicobacter and Lactobacillus. In diabetic mice, the abundance of Muribaculaceae, Lachnospiraceae, Lactobacillus and Muribaculum was significantly decreased, while Helicobacter was significantly increased. Liraglutide and empagliflozin decreased the abundance of Helicobacter and increased the abundance of Lactobacillus. In addition, empagliflozin also increased the abundance of Muribaculaceae and Muribaculum. At the same time, we also observed that the relative abundance of some bacteria changed greatly after drug treatment. The abundance of Ruminococcus was 1.39% in the DM group, but increased to 23.52% and 25.36% in the LirT and Emp&LirT groups, and 0.33% in the control group. Empagliflozin increased the abundance of Olsenella and Odoribacter.

Discussion

A variety of hypoglycemic drugs have shown protective effects on chronic diabetic kidney disease (Kuang et al., 2023), but the mechanisms involved are still unclear. Consistent with previous studies (Cha et al., 2021), this study showed that liraglutide and empagliflozin not only reduced blood glucose, but also improved diabetes-related kidney damage. Liraglutide was superior to empagliflozin at reducing serum creatinine. Empagliflozin had a better therapeutic effect in reducing serum urea nitrogen and urinary protein levels. Compared with monotherapy, combination therapy using two drugs showed greater hypoglycemic abilities. From the perspective of renal histopathological changes, both liraglutide and empagliflozin were able to alleviate renal injury and inhibit renal inflammation and fibrosis to some extent.

Lipotoxicity is not only the basic pathophysiological mechanism of diabetes, but also an important risk factor for the occurrence and development of DN. Improving lipid metabolism and reducing renal lipid accumulation are important methods used to inhibit the development of renal inflammation and fibrosis (Su et al., 2020) and other studies have shown that liraglutide can reduce ectopic lipid deposition in the renal tubules of DN rats by inhibiting SREBP-1and FAS, and increasing the expression levels of ATGL and HSL proteins. This improved the PA-induced lipid accumulation in renal tubular epithelial cells. It has been suggested that liraglutide, which is a new generation lipopeptide drug, can reduce renal lipid deposition by inhibiting lipid synthesis and promoting lipid decomposition. SGLT2 inhibitors are also known to regulate lipid metabolism (Szekeres, Toth & Szabados, 2021). Cagligin can inhibit fat synthesis by inhibiting SREBP-1c expression (Day et al., 2020) and also reduce circulating cholesterol levels in mice by inhibiting the expression of genes related to cholesterol synthesis (Osataphan et al., 2019). This study showed that both liraglutide and empagliflozin significantly decreased the levels of serum triglycerides in diabetic mice, which was consistent with the results of other previous studies (Yaribeygi et al., 2022). This is probably due to the ability of liraglutide and empagliflozin to inhibit adipogenesis via a variety of cellular pathways.

Cholesterol also an important lipid molecule, plays a central role in steroid hormone synthesis and the structure of cell membranes (Zhang & Liu, 2015). The results of this study showed that both drugs could reduce the levels of total cholesterol in serum by a small amount. The effect of liraglutide was better than that of empagliflozin, but the differences were not significant when compared with the diabetic group. For LDL, empagliflozin treatment greatly reduced serum levels in diabetic mice. This study suggests that the effect of empagliflozin in improving serum lipid metabolism may be better than that of liraglutide. However, related studies have reported that empagliflozin has different effects on cholesterol metabolism. A meta-analysis showed that SGLT2 inhibitors can increase the levels of total LDL and HDL cholesterol in T2DM patients (Sánchez-García et al., 2020). This may be related to a reduction in LDL-cholesterol clearance in circulation, along with an enhanced lipolysis of triglycerides (Basu et al., 2018). The effects and mechanisms of lipopeptide and empagliflozin on lipid metabolism still need to be further investigated.

We also observed the effects of liraglutide and empagliflozin on intestinal microflora in diabetic mice, since a number of studies have shown that there is a correlation between intestinal flora imbalance and lipid metabolism (Hu et al., 2020) as well as kidney-related indicators (Kikuchi et al., 2019). Clostridium was found to have a significant negative correlation with serum creatinine, and its presence could be used to accurately distinguish patients with DN from healthy people (Zhang et al., 2022). Improving diabetes-related renal injury through the “enterorenal axis” is the potential mechanism of many drugs, including empagliflozin. Related studies have shown that empagliflozin can restore the decrease of intestinal microflora diversity after HFD/STZ treatment, reduce the numbers of LPS-producing bacteria and Oscillibacter, and increase short-chain fatty acid (SCFA)-producing Bacteroidota and Odoribacter (Deng, Yang & Xu, 2022). It is suggested that empagliflozin can improve renal injury by remodeling the intestinal flora, and whether this is also the potential mechanism of liraglutide has not been reported.

In this study, both drugs failed to restore the alpha diversity of the flora, which is not consistent with previous results. However, a meta-analysis showed that there was no significant correlation between alpha diversity of intestinal flora and diabetes status (Gurung et al., 2020). However, there were significant changes in the structure of intestinal flora. Hand B is usually used to evaluate changes in the levels of intestinal microflora. Firstly, our data showed a significant increase in the ratio of thick-walled bacteria to Bacteroidota in the intestinal flora of diabetic mice. Previous studies have shown that an increase in the value of Fmax B reduced the production of short-chain fatty acids (SCFA) and increased energy intake, which induces non-alcoholic fatty liver disease (NAFLD) (Leung et al., 2016). Bacteroidota can carry leptin, and an increase in its abundance can reduce energy intake, thereby affecting carbohydrate fermentation and lipopolysaccharide metabolism (Sharpton et al., 2019). Other studies have shown that the decrease in its proportion is associated with obesity, which can increase with low energy intake (Ley et al., 2006). This explains the significant decrease in the abundance of intestinal Bacteroidota in diabetic mice induced a HFD in this study. Empagliflozin was able to reduce the ratio of Famp B from 7.45 to 2.04, which further confirmed that the remodeling of intestinal flora was due to this drug’s effect in regulating lipid metabolism, rather than liraglutide.

Helicobacter is thought to be associated with inflammation, lipid metabolism, and oxidative stress, and its abundance is associated with insulin resistance and obesity (Yang et al., 2023). As a traditional probiotic, Lactobacillus can alter the production of SCFA in the cecum by reducing lactic acid and increasing acetic acid production (Nguyen et al., 2022). Its related products are widely used to improve hypercholesterolemia and relieve hyperglycemia (Li et al., 2022a, 2022b; Wang et al., 2023). The data of this study showed that the abundance of Helicobacter was significantly increased in the diabetic group, and both liraglutide and empagliflozin significantly decreased the abundance of Lactobacillus and increased the abundance of beneficial bacteria. Additional difference was that empagliflozin also increased the abundance of Muribaculaceae, Muribaculum, Olsenella and Odoribacter (which was consistent with previous research (Deng, Yang & Xu, 2022)). Among these, Muribaculaceae increased insulin sensitivity, reduce host obesity by producing SCFA, promoted fat decomposition and fatty acid oxidation, and inhibited cholesterol synthesis in the liver (Bach Knudsen et al., 2018; Deehan et al., 2020).

Odoribacter can consume succinic acid in order to improve glucose tolerance and chronic inflammation which are conditions associated with diabetes (Huber-Ruano et al., 2022). Liraglutide, in particular, increased the abundance of Ruminococcus, which belongs to the phylum of thick-walled bacteria and has the ability to produce SCFA, tryptophan and bile acid metabolites (Crost et al., 2023). Its proportion is abnormally high in patients with metabolic syndrome (Grahnemo et al., 2022), although there is still no definite evidence to show whether Ruminococcus plays an active or pathogenic role in the development of the disease. Whether the increase of its abundance is the potential mechanism that positive effect of liraglutide has on DN in mice needs to be further explored.

In summary, this study showed that although the two drugs significantly changed the structure of the intestinal flora, the effects on different genera were different. In terms of improving lipid metabolism and reducing urinary protein, empagliflozin was superior to liraglutide. At the same time, it improved the intestinal community structure of DN-associated mice, which became more similar to the NC group of animals.

Conclusion

Both liraglutide and empagliflozin improved diabetes-related renal injury, but the effect of the latter on reducing LDL was better than that of the former. In addition, there was a significant difference in the effects of both drugs on the diversity and composition of the intestinal microflora.

Supplemental Information

Supplemental Information 1 Data of 16sDNA.

Supplemental Information 2 The data used for statistics in the figure.

Supplemental Information 3 ASV profiling.

Supplemental Information 4 Author checklist.

We are grateful to Dr. Dev Sooranna of Imperial College London for English writing guidance.

Additional Information and Declarations

Competing Interests

Author Contributions

Animal Ethics

DNA Deposition

Data Availability

The authors declare that they have no competing interests.

Qiong Yang conceived and designed the experiments, analyzed the data, authored or reviewed drafts of the article, and approved the final draft.

Ling Deng conceived and designed the experiments, performed the experiments, prepared figures and/or tables, and approved the final draft.

Changmei Feng performed the experiments, prepared figures and/or tables, and approved the final draft.

Jianghua Wen analyzed the data, authored or reviewed drafts of the article, and approved the final draft.

The following information was supplied relating to ethical approvals (i.e., approving body and any reference numbers):

The experimental protocols of all mice were approved by the Animal Experiment Ethics Committee of Guangxi Medical University (NO.20202014).

The following information was supplied regarding the deposition of DNA sequences:

The sequences are available at Sequence Read Archive (SRA): PRJNA1017836.

https://www.ncbi.nlm.nih.gov/sra/PRJNA1017836.

The following information was supplied regarding data availability:

The raw measurements are available in the Supplemental Files.

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
