# Peer review of "Comparing the effects of empagliflozin and liraglutide on lipid metabolism and intestinal microflora in diabetic mice"

_PeerJ, doi:10.7717/peerj.17055_

## Round 0.1 · original submission · Minor Revisions

Dear authors, thank you for your submission. At this stage, your manuscript requires minor to medium revisions in order for us to move forward. Please, refer to the reviewers comments for more details. Additionally, please explain what is the collagen graph (no pixel quality) under the kidney histology images, figure 1. Also, improve and complete your figures legends but indicating major findings or key highlights probably identified (e.g. using * or letters in the histology images identifying the disorder, etc). Identify in your methods the suppliers and reference numbers of antibodies, ELISA kits used, etc. .. remember that if wanted, any other researchers should be able to reproduce your experiments and thus this will help with transparency and completeness. Information on protein content loaded, amount of DNA used, etc should also be provided. Therefore, please revise and Proofread your manuscript throughout. Also, improve your statistics description.

**Language Note:** The Academic Editor has identified that the English language must be improved. PeerJ can provide language editing services - please contact us at copyediting@peerj.com for pricing (be sure to provide your manuscript number and title). Alternatively, you should make your own arrangements to improve the language quality and provide details in your response letter. – PeerJ Staff

We suggest editing the title to:

> Comparing the effects of empagliflozin and liraglutide on lipid metabolism and the intestinal microflora in diabetic mice

or

> The effects of empagliflozin and liraglutide on lipid metabolism and the intestinal microflora in diabetic mice

Reviewer 1 ·

Basic reporting

1.“In addition, our data suggested that liraglutide may still be better than in reducing cholesterol levels:,“Because they are far away from each group”et.al,these sentences are not clear in article.
2.Paragraphings are too frequent and there is a lack of words between paragraph.

Experimental design

“all treatments lasted for 8 weeks.”which group are dealt with in this way, and what about the others?

Validity of the findings

1.“However, Bacteroidota faecalis and Clostridiumspp. have been shown to significantly enrich in fecal samples of DN patients”uncited references.

Additional comments

1.The right side line is not aligned.

·

Basic reporting

This paper compares the effect of empagliflozin and liraglutide on kidney, lipid metabolism and the intestinal microbiota in diabetic mice. The paper is well written and structured.

Experimental design

This is a well-done study. The logical flow of the experiments is good, and the work appears to have been done well. To understand the impact of two widely used anti-diabetic drugs on the kidney, lipid metabolism and the intestinal microflora is crucial. I do not have any major criticisms.

Validity of the findings

Using a range of techniques, the authors were able to determine that both drugs were associated with improvement of all indexes related to renal function. However, the effect in the intestinal microflora were distinct.

Minor comments:

The legends are at times incomplete. For example, they should describe the type of graph used to show the data (i.e. bar graph, violin plot etc.). And also, it should be specified the type of analyses (i.e. t-test, One-Way ANOVA etc.). In the methods, it is unclear which protocol has been used for the Masson staining.

Reviewer 3 ·

Basic reporting

In the submitted manuscript by Yang et al, the authors compared the effect of two drugs empagliflozin and liraglutide in diabetic mice including the lipid metabolism and the intestinal microflora. The results nicely showed that both drugs have positive effect on renal function, lipid metabolism, however, small difference remains between the two drugs. Moreover, they found that the drugs can modulate the composition and diversity of microbiota from the fecal samples of diabetic mice.
In general, the studies were well performed and results thoroughly discussed. However, the manuscript can still benefit from the language improve and many typos in the context. Please take time in working on it.
Is there any sample from the patients that treated with those two drugs? It would be nice to have a parallel comparison between mice and human.
FIgure1: an extra graph in the bottom not annotated. Could you show individual samples in figure 1B-D like it is shown in figure 1A.

Experimental design

N/A

Validity of the findings

N.A

Additional comments

NA

Reviewer 4 ·

Basic reporting

Review for Yang et al “Compare the effects of Empagliflozin and Liraglutide
on lipid metabolism and the intestinal microflora in diabetic mice”

This study delves into the comparative impacts of empagliflozin and liraglutide, two commonly used antidiabetic medications, on kidney function, lipid metabolism, and intestinal microbiota in diabetic mice. Using a type 2 diabetes mouse model induced by a high-fat diet and streptozotocin injection, the research explores how these drugs affect various health parameters. Both empagliflozin and liraglutide demonstrated significant improvements in renal function, but empagliflozin outperformed in reducing low-density lipoproteins (LDL), while liraglutide showed better efficacy in reducing serum cholesterol. Interestingly, the drugs exerted distinct influences on the intestinal flora: empagliflozin notably altered the Firmicutes to Bacteroidota ratio and increased the abundance of specific bacteria, such as Muribaculaceae, Muribaculum, Olsenella, and Odoribacter. In contrast, liraglutide increased the abundance of Ruminococcus. This study underscores the nuanced effects of these medications, highlighting the importance of tailoring treatments for individuals managing diabetes. Ultimately, both drugs exhibited positive outcomes in addressing diabetes-related renal issues, with empagliflozin potentially standing out for its LDL reduction capabilities.

The manuscript has many typographical errors and needs significant improvements. However, the experiments are well executed with rigor.

My concerns-

Line 20 It should be "inhibitor" instead of "inhibiter"

Line 24 “The mouse model of type 2 diabetes was established by feeding rats a high fat
diet” you used mice or rats please clarify.

Line 35 Abbreviate empagliflozin and other terms consistently in abstract.

Line 50 Ensure consistency in the citation format “(Alicic, Rooney, and Tuttle 2017)” see
if it matches the format with other in papers.

Line 95 Give information about protein content in the diet.

Line 97 Give right STZ dose unit 90mg/k gip.

Line 98 What was the frequency of STZ, was it daily for 5 consecutive days or a single
dose. Clarify in methods.

Line 104 Do not give full form when you have abbreviated previously for diabetic model,
liraglutide treatment for rest of the text also.

Line 129 Give speed and duration of centrifugation.

Line 129 Provide more details about the duration of urine collection, storage conditions,
and the specific measurements conducted with the automatic biochemical
analyzer.

Line 139 Specify the staining protocols more precisely. For example, provide
information on the specific solutions used for fixation, dehydration, and
embedding. Mention the staining duration and conditions for oil red, HE,
PAS, Masson, and immunohistochemical staining.

Line 144 How DNA isolation was done. If you have used kit, give details.

Line 170 correct spelling of “Kruskal-Walis test” to “Kruskal-Wallis test”.

Suggestion for discussion
Discuss the translational potential of your findings. Consider how the observed effects in mice may or may not translate to human patients and discuss the relevance of your results to the clinical management of diabetes and associated complications.

Experimental design

nothing to add

Validity of the findings

nothing to add

---

## Round 0.2 · accepted · Accept

I am happy to let you know that your manuscript has now been approved for publication. Congratulations and thank you for your hard work.

·

Basic reporting

The comments have been addressed.

Experimental design

The comments have been addressed.

Validity of the findings

The comments have been addressed

Reviewer 3 ·

Basic reporting

N/A

Experimental design

N/A

Validity of the findings

N/A

Additional comments

I appreciate the authors' detailed point-to-point response to my previous comments. I look forward to the final version of the manuscript. Congrats!

Reviewer 4 ·

Basic reporting

'no comment'

Experimental design

'no comment

Validity of the findings

'no comment

Additional comments

I appreciate the authors for their prompt and thorough revisions addressing my earlier concerns. The modifications have significantly strengthened the manuscript, and I am now satisfied with the completeness and clarity of the content.